# Genotype–Trait (GT) Biplot Analysis for Yield and Quality Stability in Some Sweet Corn (*Zea mays* L. *saccharata* Sturt.) Genotypes

Atom Atanasio Ladu Stansluos [1], Ali Öztürk [2], Gniewko Niedbała [3,*], Aras Türkoğlu [4,*], Kamil Haliloğlu [2], Piotr Szulc [5], Ali Omrani [6], Tomasz Wojciechowski [3] and Magdalena Piekutowska [7]

1    Department of Field Crops, Faculty of Agriculture, Upper Nile University, Malakal 71100, South Sudan; atomtaban@gmail.com
2    Department of Field Crops, Faculty of Agriculture, Ataturk University, 25240 Erzurum, Turkey; aozturk@atauni.edu.tr (A.Ö.); kamilh@atauni.edu.tr (K.H.)
3    Department of Biosystems Engineering, Faculty of Environmental and Mechanical Engineering, Poznań University of Life Sciences, Wojska Polskiego 50, 60-627 Poznań, Poland; tomasz.wojciechowski@up.poznan.pl
4    Department of Field Crops, Faculty of Agriculture, Necmettin Erbakan University, 42310 Konya, Turkey
5    Department of Agronomy, Poznań University of Life Sciences, Dojazd 11, 60-632 Poznań, Poland; piotr.szulc@up.poznan.pl
6    Crop and Horticultural Science Research Department, Ardabil Agricultural and Natural Resources Research and Education Center, AREEO, Moghan 193951113, Iran; a.omrani@areeo.ac.ir
7    Department of Geoecology and Geoinformation, Institute of Biology and Earth Sciences, Pomeranian University in Słupsk, 27 Partyzantów St., 76-200 Słupsk, Poland; magdalena.piekutowska@apsl.edu.pl
*    Correspondence: gniewko.niedbala@up.poznan.pl (G.N.); aras.turkoglu@erbakan.edu.tr (A.T.)

**Abstract:** A strong statistical method for investigating the correlations between traits, assessing genotypes based on numerous traits, and finding individuals who excel in particular traits is genotype–trait (GT) biplot analysis. The current study was applied to evaluate 11 sweet corn (*Zea mays* L. *saccharata*) genotypes and correlate them based on genotype–trait (GT) biplot analysis for two cropping seasons in Erzurum, Türkiye using the RCBD experimental design with three reputations. The results showed that the genotypes were significantly different for the majority of the examined variables according to the combined analysis of variance findings at 0.01 probability level. An ecological analysis was performed to evaluate sweet corn varieties and environmental conditions and interactions between them (genotype × environmental conditions). Our results showed that the summation of the first two and second main components was responsible for 73.51% of the combined cropping years of the sweet corn growth and development variance, demonstrating the biplot graph's optimum relative validity, which was obtained. In this study, the Khan F1 (G6) genotype was found to be the stablest genotype, and the Kompozit Seker (G7) genotype was the non-stable genotype, moreover based on the first cropping year, second cropping year, and the average mean of the two cropping years. As a conclusion, the Khan F1 (G6) genotype is the highest-yielding genotype, and the Kompozit Seker (G7) is the lowest. Based on the heat map dendrogram, the context of the differential extent of trait association of all genotypes into two clusters is indicated. The highest genetic distance was shown between the BATEM Tatlı (G3) and Febris (G5) genotypes. Our results provide helpful information about the sweet corn genotypes and environments for future breeding programs.

**Keywords:** GGE biplot methods; genotype × environment interaction; stability; sweet corn; heat map dendrogram

## 1. Introduction

The annual world harvested area of sweet corn is 1,042,894.0 hectares, production is 8,858,138.9 tons, and yield is 8493.8 kg/ha. Currently, the biggest producers of sweet

corn are the USA (2,617,864.0 tons), Mexico (1,059,259.9 tons), Nigeria (775,989.6 tons), and Indonesia (653,821.9 tons) [1]. Despite this increase in sweet corn production, since the increasing needs of the relevant sectors could not be met, Türkiye imported 8.02 million tons of corn and its products in 2019, 2020, and 2021 against a total of 2.5 million tons of exports [2]. Therefore, increasing the production of high-yielding varieties with optimum and stable kernel yield is requested.

Sweet corn grains are rich in tocopherols, carotenoids, vitamins, and phenolic compounds [3]; Sweet is superior to other genotype groups in terms of sugar, protein, fat, some vitamins (A, B, C, E, and K) and nutritional elements (Ca, K, Fe, Na, and Zn) [4]. Sweet corn cultivation areas are gradually increasing in Türkiye, contract production is carried out for the canned and frozen product industry, and it is estimated that the sweet corn cultivation area is 1–2% of the total maize cultivation area [4,5]. Varieties with high yield and quality adaptability to the ecological conditions of the region should be evaluated.

The short period of the rainy season and the limited amount of rain showed the importance of adaptation experiments in selecting and improving the appropriate genotypes with high yield potentials within the existing gene pool conditions through a wide range of environments. Because of the nutritional advantage of maize protein quality (QPM), the improvement of extremely early-harvesting QPM genotypes for kernel yield to provide for the quality needs of the growing population to challenge climate change and meet their daily nutritive requirement [6] is needed. For positive and effective results of the adaptation, the effect of the environmental conditions on the genotypes is one of the important factors [7].

The interaction of genotype × environment which mostly exists due to one or more of the environmental traits such as rainfall amount and distribution, years, and seasons [4] may help breeders to evaluate the yielding of the genotype in a certain environment and determine the optimum yield of the genotype(s) [8]. The yielding of hybrids and their reaction to the environment can be evaluated using several statistical methods, including parametric and non-parametric. The essential multivariate methods that are used for data analysis of the yield comparison research studies are such statistical methods as genotype and genotype by environment (GGE), biplot additive main effects and multiplicative interaction (AMMI), and principal component analysis (PCA). These methods are used to estimate the sustainable stability and compatibility of the genotypes through a quantitative analysis of singular value decomposition [9,10]. It was confirmed that the GGE biplot methodology is more fruitful than the AMMI methodology in analyzing the genotype data in different environmental conditions. Many scholars presented the GGE biplot methodology as an effective tool for evaluating the interaction between the genotype and the environmental conditions (genotype × environment). The genotype and the genotype-by-environment (GGE) biplot method is an influential analytical means of evaluating the interaction between the genotype and the environmental conditions [11]. To evaluate the genotypes in different environmental conditions, the environmental effect has been elaborated considerably in more studies. Therefore, eliminating the environmental effect and only concentrating on the key effects is necessary for the genotypes (G) and their interaction with the environmental conditions (GE) [12]. The summation of genotypes and genotype effects within a certain environment is significant for the selection of stable genotypes. Therefore, the evaluation of the genotype effects and their interaction with the environmental conditions contemporaneously is remarkable. The GGE biplot methodology allows the effect of the genotypes and the environmental conditions to be estimated concurrently and credibly [13], as recommended by Yan et al. [10]. The GGE biplot graphical method can play the final role in the selection of genotypes and the genotypes' environments [13,14]. Therefore, GGE biplot analysis is superior to other statistical methodologies due to its efficiency in explaining the sum of squares of GE and G + GE and due to its superior predictive accuracy in maize genotypes, and it is also a valuable method of identifying environmental conditions which can differentiate between genotypes and can be used to identify promising maize genotypes in such environmental conditions [15,16]. The genotype–trait (GT) interaction

is one of the GGE biplot methods of studying genotype–trait interaction data, and this research proposed that GT biplot is an appropriate tool for identifying cultivars and trait interactions; in this analysis, genotypes are considered as lines and traits as testers [17]. Selecting various attributes with high performance and success can be achieved using the GT biplot approach [7,18]. The correlation of traits, as determined by the GGE biplot approach, is shown on genotype–trait biplot diagrams [19]. Swelam in 2012 [8] and Yan in 2014 [11] looked at the connections between genotype and traits using GT biplot diagrams. Genotype–trait biplot diagrams were employed by Adedeji et al. [20], Yan [21], and Shojaei et al. [7] to investigate the connections between genotypes and characteristics. Although there is little research on sweet corn using GT biplot analysis, the current study's objectives include determining the genotypes that had the most desirable traits for various traits, examining the correlation between the studied traits and their relationships, and categorizing genotypes in accordance with the studied traits. This is achieved by using the GT biplot method to investigate the effect of genotype traits.

## 2. Materials and Methods

### 2.1. Experimental Specifications and Soil Sample Analyses

Eleven sweet corn genotypes (Table 1) were evaluated for yield and some quality characteristics in the two previous papers. Genetic groups and the other agronomic characteristics of the genotypes used in the research, details of the environmental conditions, and experimental design were presented in these papers [22,23]. The experiments were applied at Ataturk University Plant Production Application and Research Centre during the 2017 and 2018 cropping seasons in Erzurum, Türkiye. Erzurum is located in the northeast part of Türkiye at 39°55′ north latitude and 41°61′ east longitude. It has land with terrestrial climate and an elevation of 1853 m above sea level. Winter is usually snowy and cold, and summer is cool and dry. The soil characteristics of the cultivated land were clay–loam, pH of 7.7, lime 5.6%, total nitrogen 0.09%, and organic matter 1.7%. The soil texture was characterized as 67% clay, 27% silt, and 36% sand. All the meteorological characteristics during the plant growth period is presented in Figure 1. The size of each plot was 2.5 m × 5.0 m, and each plot had five plant rows, giving a total plot size of 12.5 m$^2$. The agricultural practices were managed according to the recommendations of [22,23]. During the sowing process, the intra-row spacing of the plants was 25 cm, while the inter-row spacing was 50 cm.

**Table 1.** Seed source and names and code of sweet corn varieties studied in the project.

| Genotype no. | Genotype | The Institution | Characteristics |
|---|---|---|---|
| G1 | Argos | Fito Tohumculuk Tic. Ltd. Şti. Antalya | Super-sweet, maturity period 80–90 days, kernel color is yellowish golden, tolerant of transportation |
| G2 | Baron F1 | May Tohum Ltd. Şti. Bursa | Super-sweet, very early, plant height 190–195 cm, ear length 19.9 cm, ear diameter 5.3 cm, number of rows per ear 16–18, unhusked ear weight 330–335 g, kernel color is dark yellow, tolerant to lodging, non-stick on the tooth, tolerant to transportation |
| G3 | BATEM Tatlı | Karya Tarım Tic. A.Ş. Aydın | Normal sweet, maturity period 70–80 days, total sugar content 13.8–15.3%, crude protein content 13.7–13.9% |
| G4 | Challenger | Monsanto Gıda ve Tarım Tic. Ltd. Şti. Bursa | Super-sweet, maturity period 80–85 days, high sugar content, kernel color is yellow, plant height 170–180 cm |
| G5 | Febris | Fito Tohumculuk Tic. Ltd. Şti. Antalya | Super-sweet, maturity period 87 days, ear length 16–18 cm, tolerant to lodging, medium height |
| G6 | Khan F1 | May Tohum Ltd. Şti. Bursa | Super-sweet, early, high yielding, plant height 190–200 cm, ear length 22–23 cm, ear diameter 5–5.2 cm, number of rows per ear 16–18, unhusked ear weight 340–350 g, kernel color is dark yellow, tolerant to lodging, tolerant to transportation |

| Genotype no. | Genotype | The Institution | Characteristics |
|---|---|---|---|
| G7 | Kompozit Şeker | Mısır Araştırma Enst. Müd. Sakarya | Normal sweet, maturity period 77–84 days, plant height 80–220 cm, fresh ear yield 12,500–21,000 kg/ha, kernel color is orange |
| G8 | Overland | Alfa Tohum Tarım Gıda İnş. Hayv. Paz. San. Tic. Ltd. Şti. Balıkesir | Normal sweet, days to flowering 65–69 days, plant height 200–220 cm, ear length 16–20 cm, number of rows per ear 18–20, kernel color is white, ear yield 8500–10,500 kg/ha, sugar content 3.8%, protein content 11.3% |
| G9 | SHY1036 | Monsanto Gıda ve Tarım Tic. Ltd. Şti. Bursa | Super-sweet, tolerant to diseases, maturity period 100–110 days, high sugar content, kernel color is yellow, plant height 210–220 cm |
| G10 | Signet | Monsanto Gıda ve Tarım Tic. Ltd. Şti. Bursa | Sugary enhanced, maturity period 60–65 days, high sugar content, kernel color is yellow, plant height 150–160 cm |
| G11 | Tanem F1 | May Tohum Ltd. Şti. Bursa | Normal sweet, early, plant height 170–180 cm, first ear height 45–60 cm, ear length 20–22 cm, ear diameter 4.5–5 cm, number of rows per ear 16–18, ear weight 460–480 g, kernel color is yellow |

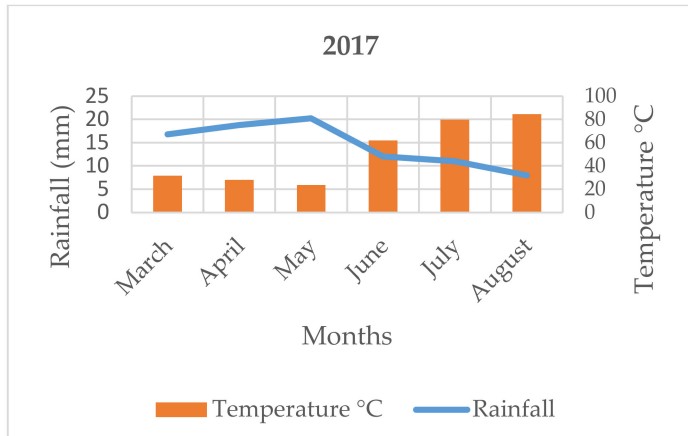
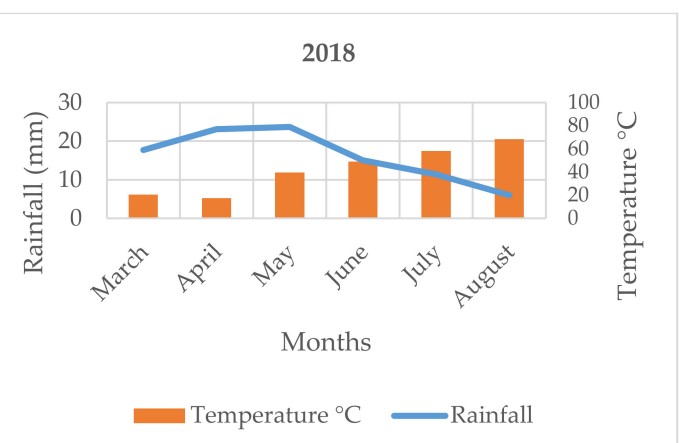

**Figure 1.** Average meteorological characteristics during the plant growth period in the experimental years.

### 2.2. Different Characteristics of Sweet Corn Genotypes

Days to silking, days to maturity, plant height, number of tillers per plant, number of leaves per plant, number of ears per plant, first ear height, number of plants per hectare, ear yield, number of marketable ears per hectare, the yield of the marketable ear, number of kernel rows per ear, number of kernels per row, number of kernels per ear, fresh kernel yield, and green mass yield were stated in related methods in each plot. Days to maturity were counted starting from the sowing date until the kernel moisture reached $73 \pm 1\%$. Marketable ears were the ones with at least 15 cm of ear length and 3 cm of ear diameter [22]. Ear length, ear diameter, ear weight, and 1000-kernel weight were calculated from the ten ears randomly selected from the marketable ears. Grain N percentages were measured using the micro-Kjeldahl procedure, and grain protein content was calculated as $6.25 \times$ percent N in dry matter. The total soluble solid content was measured by using a digital hand refractometer at harvest—seven days post-harvest for ears with husk which were stored in the refrigerator and for ears with harvest dates delayed one week at the field. Grain moisture content was determined as a percentage seven days post-harvest in the refrigerator and field [24].

### 2.3. Data Calculation and Analysis

In this research study, graphical decomposition was performed using a GT biplot according to each quantity by the following equation:

$$\frac{\alpha_{ij} - \beta_j}{\sigma_j} = \sum_{n=1}^{2} \lambda_n \xi_{in} \eta_{jn} + \varepsilon_{ij} = \sum_{n=1}^{2} \xi_{in}^* \eta_{jn}^* + \varepsilon_{ij}$$

where $\alpha_{ij}$: average amount of genotype *i* for every trait *j*; $\beta_j$: average amount of all the genotypes for the traits; $\sigma_j$: standard deviation of the trait *j* in the average genotypes; $\varepsilon_{ij}$: amount of genotype *i* remained in the trait j; $\lambda_n$: certain amount for the main element (PCn); $\xi i$: amount of PCn for the genotype *i*; $\eta_{jn}$: amount of PCn for the genotype *j* [7]. Four GT biplot diagrams (ranking and grouping of genotypes based on attributes, ranking of genotypes based on performance stability, and ranking of genotypes) were utilized in this work based on the optimum genotype and classification of genotypes according to the investigated attributes. Heat maps based on genetic and phenotypic correlations were used to investigate relationships between the studied traits. SAS.v 9.2 software was used in the statistical analyses. Analysis of variance, comparison of means via the Duncan method, correlation coefficients between traits, and Genstat.V12 software were also used to graphical analysis in each environment.

### 3. Results and Discussion

### 3.1. Analysis of Variance and Mean Comparison

The results of variance analysis of the data indicated that the genotypes had very significant differences in terms of most of the traits (Table 2). The highest plant height, number of tillers per plant, first ear height, fresh kernel yield, grain protein content, total soluble solid content at harvest, and total soluble solid content at harvest seven days post-harvest in the field values were obtained in 2017, while the highest number of leaves per plant, ear length, ear diameter, number of kernels per row, number of kernels per ear, grain moisture content at seven days post-harvest in the field, and grain moisture content at seven days post-harvest in the refrigerator were obtained in 2018. Among the genotypes under this study; Baron F1 showed the best performance in terms of ear diameter and ear weight; BATEM Tatlı for days to silking, days to maturity, first ear height, and grain protein content; Challenger for number of marketable ears per hectare; Febris for number of leaves per plant; Khan F1 for ear length, number of kernels per row, grain moisture content at seven days post-harvest in the field, and grain moisture content at seven days post-harvest in the refrigerator; Kompozit Şeker for plant height and number of plants per hectare; SHY1036 for number of kernel rows per ear, number of kernels per ear, and green mass yield; Signet for number of tillers per plant, number of ears per plant, ear yield, the yield of the marketable ear, 1000-kernel weight, and fresh kernel yield; and Tanem F1 for total soluble solid content at harvest, total soluble solid content at harvest seven days post-harvest in the field, and total soluble solid content seven days post-harvest in the refrigerator [22,23]. Considering that the FKY trait is one of the most widely used and important traits in genotype × trait effect experiments, similarly, the comparison analysis of the means effect of year × genotype was used to investigate different genotypes in the years of the experiment. Based on this diagram, the Challenger genotype in the second year of the experiment and the Signet genotype in the first and second years of the experiment had high favorability in terms of FKY traits. Additionally, the Kompozit Şeker and BATEM Tatlı genotypes were identified as unfavorable genotypes in terms of this trait in both years of the experiment (Figure 2).

**Table 2.** Combined analysis of variance for growth, yield components, yields, and some quality characteristics of 11 sweet corn genotypes grown under Erzurum conditions in the 2017 and 2018 crop seasons.

| Trait No. | Trait | Genotype Effect (*p* Value) | Year Effect (*p*) | Genotype × Year Effect (*p*) | CV (%) |
|---|---|---|---|---|---|
| T1 | Days to silking (DS) | 0.180 | 0.000 | 0.000 | 1.67 |
| T2 | Days to maturity (DM) | 0.090 | 0.000 | 0.051 | 1.64 |
| T3 | Plant height (PH) | 0.000 | 0.000 | 0.476 | 4.89 |
| T4 | Number of tillers per plant (NTP) | 0.000 | 0.000 | 0.000 | 22.81 |
| T5 | Number of leaves per plant (NLP) | 0.000 | 0.000 | 0.007 | 6.97 |
| T6 | Number of ears per plant (NEP) | 0.688 | 0.000 | 0.041 | 5.75 |
| T7 | First ear height (FEH) | 0.000 | 0.000 | 0.019 | 7.53 |
| T8 | Number of plants per hectare (NPH) | 0.052 | 0.000 | 0.999 | 6.58 |
| T9 | Ear yield (EY) | 0.352 | 0.000 | 0.602 | 9.97 |
| T10 | Number of marketable ears per hectare (NMEH) | 0.886 | 0.000 | 0.935 | 12.89 |
| T11 | The yield of the marketable ear (YME) | 0.713 | 0.000 | 0.010 | 10.93 |
| T12 | Ear length (EL) | 0.032 | 0.000 | 0.050 | 3.47 |
| T13 | Ear diameter (ED) | 0.005 | 0.000 | 0.000 | 3.07 |
| T14 | Ear weight (EW) | 0.449 | 0.000 | 0.004 | 12.57 |
| T15 | Number of kernel rows per ear (NKRE) | 0.177 | 0.000 | 0.328 | 3.98 |
| T16 | Number of kernels per row (NKR) | 0.000 | 0.000 | 0.000 | 3.86 |
| T17 | Number of kernels per ear (NKE) | 0.003 | 0.000 | 0.083 | 5.50 |
| T18 | 1000-kernel weight (TKW) | 0.456 | 0.000 | 0.000 | 5.05 |
| T19 | Fresh kernel yield (FKY) | 0.024 | 0.000 | 0.032 | 11.34 |
| T20 | Grain protein content (GPC) | 0.000 | 0.000 | 0.000 | 7.13 |
| T21 | Total soluble solid content at harvest (SSCH) | 0.666 | 0.000 | 0.000 | 7.65 |
| T22 | Grain moisture content at seven days post-harvest in the field (GMCF) | 0.006 | 0.000 | 0.706 | 2.37 |
| T23 | Grain moisture content at seven days post-harvest in the refrigerator (GMCR) | 0.012 | 0.000 | 0.975 | 2.67 |
| T24 | The total soluble solid content at harvest seven days post-harvest in the field (SSCF) | 0.000 | 0.000 | 0.012 | 8.53 |
| T25 | Total soluble solid content seven days post-harvest in the refrigerator (SSCR) | 0.016 | 0.000 | 0.018 | 10.04 |
| T26 | Green mass yield (GMY) | 0.275 | 0.000 | 0.477 | 9.21 |

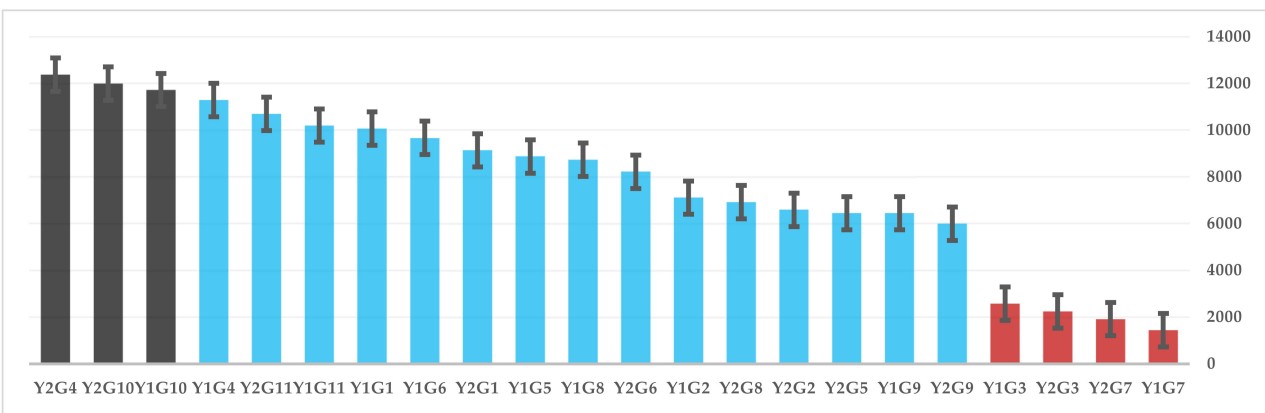

**Figure 2.** Comparison of year × genotype effect in terms of FKY trait in two years of experiment on 11 maize genotypes.

### 3.2. Ranking and Grouping of Genotypes in Terms of Traits

Considering the contemporaneous study research of genotypes performance and sustainable stability can gain the biplot diagram of average environmental coordinates. These diagrams are used for examining the performance and genotypes' sustainable stability under that environment. A biplot diagram of average environmental coordinates in the biplot GGE methodology is one of the suitable methodologies in stability analysis [25]. A polygon diagram demonstrates the ultimate genotypes through the traits under this study. This diagram is demarcated by relating the genotypes outermost from the origin so that the rest of the genotypes can fit into that polygon. In each fragment, genotypes with high-yielding performance and desirability with a particular characteristic are distinguished by lines [26,27]. Dolatabad et al. [26] used this type of diagram in their research study on rapeseed genotypes and maize genotypes.

For the polygon diagram in year one (Figure 3A), the first principal component covers more than 46%, the second principal component covers more than 22%, and a total of 68.44% of the data variance is covered by these two components. The following genotypes had the farthest distance from the origin of the diagram: BATEM Tatlı, Febris, Kompozit Seker, SHY1036, and Signet. They were stated at the polygons' apex. Names of the appropriate genotypes were acknowledged in terms of the studied characteristics. In every fragment, the BATEM Tatlı genotype in terms of the crude protein (CP%) and total soluble solids 7 days after harvest in the field (SSCF); the BATEM Tatlı genotypes in terms of number of kernel rows per ear (NKRE) and number of kernels per ear (NKE); the KOMPOZIT ŞEKER genotype in terms of number of plants per hectare (NP/ha); the SHY1036 genotype in terms of days to harvest (DM); and the Signet genotype in terms of number of tillers per plant (NTP), number of ears per plant (NEP), and 1000-kernel weight (TKW) were recognized as more suitable than the other genotypes (Figure 3A).

Based on the polygon diagram in the second year (Figure 3B), the first principal component covers more than 48%, the second principal component covers more than 21%, and a total of 70.42% of the data variance is covered by these two components. Based on this, the Argos, BATEM Tatlı, Kompozit Seker, SHY1036, Signet, and Tanem F1 genotypes had the farthest distance from the origin of the diagram. They were instated at the apex of the polygon. The names of appropriate genotypes were identified in terms of characteristics. In every fragment, the Argos genotype in terms of the number of ears per plant (NEP), the Kompozit Şeker genotype regarding the number of plants per hectare at harvest (NP/ha), the SHY1036 genotype regarding the number of leaves per plant (NLP), the Signet genotype regarding the 100-kernel weight (TKW), and the Tanem F1 genotype regarding the total soluble solids (SSCF) were identified as more successful hybrids than other genotypes (Figure 3B).

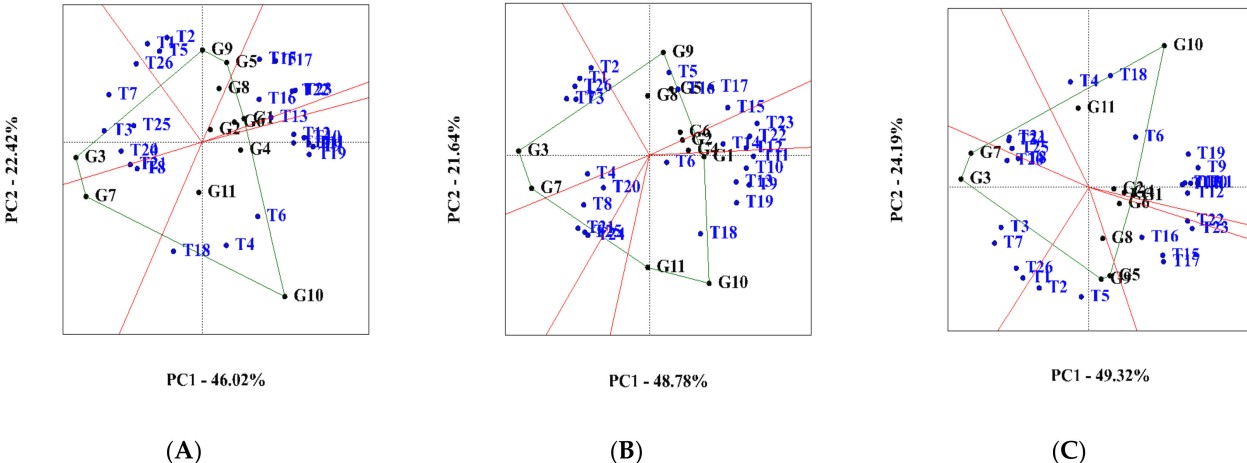

**Figure 3.** Ranking and grouping of genotypes in terms of genotype × traits. (**A**): First cropping year, (**B**): second cropping year, (**C**): average of two cropping years. G1: Argos, G2: Baron F1, G3: BATEM Tatlı, G4: Challenger, G5: Febris, G6: Khan F1, G7: Kompozit Şeker, G8: Overland, G9: SHY1036, G10: Signet, and G11: Tanem F1. T1: DS, T2: DM, T3: PH, T4: NTP, T5: NLP, T6: NEP, T7: FEH, T8: NPH, T9: EY, T10: NME, T11: YME, T12: EL, T13: ED, T14: EW, T15: NKRE, T16: NKR, T17: NKE, T18: TKW, T19: FKY, T20: GCP, T21: SSCH, T22: GMCF, T23: GMCR, T24: SSCF, T25: SSCR, and T26: GMY.

Based on the polygon diagram combined for the two years (Figure 3C), the first principal component covers more than 49%, the second principal component covers more than 24%, and a total of 73.51% of the data variance is covered by these two components. Accordingly, the Argos, BATEM Tatlı, Febris, Kompozit Seker, SHY1036, and Signet genotypes had the farthest distance from the origin of the diagram. They were placed at the apex of the polygon. The names of desirable genotypes were identified regarding the characteristics. In every fragment, the Argos genotype regarding the number of ears per plant (NEP); the Kompozit Seker genotype regarding crude protein (CP%), total soluble solids at harvest (SSCH), total soluble solids 7 days after harvest in the field (SSCF), and total soluble solids 7 days after harvest in the refrigerator (SSCR); and the SHY1036 genotype in terms of number of leaves per plant (NLP), SIGNET 100 kernel weight (TKW), and the number of tillers per plant (NTP) were identified as more suitable genotypes than the others (Figure 3C).

Comparing the first and second environmental conditions, it can be stated that based on this diagram, BATEM Tatlı, Kompozit Seker, Signet, and SHY1036 genotypes are identified as desirable genotypes. In the case of the studied traits, the number of leaves per plant (NLP) shows great sustainable stability and performance. Various researchers used polygon diagrams in order to investigate corn genotypes and evaluated traits in terms of the degree of desirability in each genotype in different years and chose this type of graph as a very widely used analysis in separating and evaluating genotypes in terms of different traits [4,7,28,29]. This genotype had a higher responsiveness to the stimuli of this set of traits because it was at the vertex of the polygon containing all the different traits [30]. In contrast, genotypes that produce polygon vertices but do not exhibit any clustered traits are viewed negatively for the various traits examined, revealing low responsiveness and yield. A variety of tools are available with the GGE biplot approach to efficiently select genotypes. The importance of the GGE biplot's effect demonstrated that the environments (first and second years) can be categorized based on interaction. It might be because of the interaction between factors such as genotype, year, and their interactions. Our findings are supported by Ma et al. [31], Mafouasson et al. [32], and Shojaei et al. [29].

### 3.3. Ranking of Genotypes × Traits in Terms of Stability in Performance

Graphic analysis was used to determine and interpret the difference between genotypes and environmental conditions. Within the axis (horizontal) PC1 biplot diagram, the key effect of the genotype and the axis (vertical) PC2 represent interactions between the

genotypes and environments, which determines the instability of the genotypes. Genotypes that are near the origin of the axis are more stable than those far from the origin of the axis. Depending on the ultimate genotype diagram, Baron F1 was determined to be the nearest genotype to the positive end of the year one cropping season, and Signet was determined as the most far genotype from the positive end. Therefore, these genotypes were identified as non-favorable genotypes (Figure 4). The order of genotypes is as follows: Baron F1 > Khan F1 > Argos > Overland > Febris > Tanem F1 > Challenger > SHY1036 > Kompozit Seker (G7) > BATEM Tatlı > Signet (Figure 4A). Concentrating on the biplot results of the year two cropping season, Khan F1 was the superlative genotype and Signet was the favorable genotype (Figure 4A). The ranking of genotypes in the year two cropping season is as follows: Khan F1 > Baron F1 > Challenger > Argos > Febris > Kompozit Seker > Overland > BATEM Tatlı > SHY1036 > Tanem F1: Tanem F1 > Signet (Figure 4B). Biplot results that were obtained from the average data of the year one cropping season and year two cropping season show that the Khan F1 is named as the superior genotype and Signet as the non-favorable genotype. The genotypes ordered based on the ideal genotype as follows: Khan F1 > Baron F1 > Challenger > Argos > Overland > Kompozit Seker > Febris > BATEM Tatlı > SHY1036 > Tanem F1 > Signet (Figure 4C). The genotypes can be grown in each of the identified environments in which they showed a comparative advantage in grain yield in this study. The genotypes that did not fit into any of the environments, however, suggest that they are not appropriate for cultivation in any of the environments taken into account in this study. In plant breeding studies, the biplot ranking model has been utilized as a useful feature to aid in visual comparison and evaluate the stability and adaptability of genotypes [33]. Based on the ranking of genotypes for performance stability in the first and second years, as well as the two years taken together, PC1 was found to be superior to PC2 (Figure 4A–C). An ideal genotype reportedly had a high PC1 score and consistent performance across environments with a low PC2 score [34,35]. Our results were confirmed by Akinyosoye [6] and Shojaei et al. [7], who used this diagram to evaluate genotypes and traits, and based on this diagram, they chose the most stable genotypes. Illés et al. [36] used the Ranking of genotypes in terms of stability in the performance diagram to investigate the phenology and performance indicators of sweet corn genotypes and, as a result, declared that the first principal component covered more than 73% of the variance of the data.

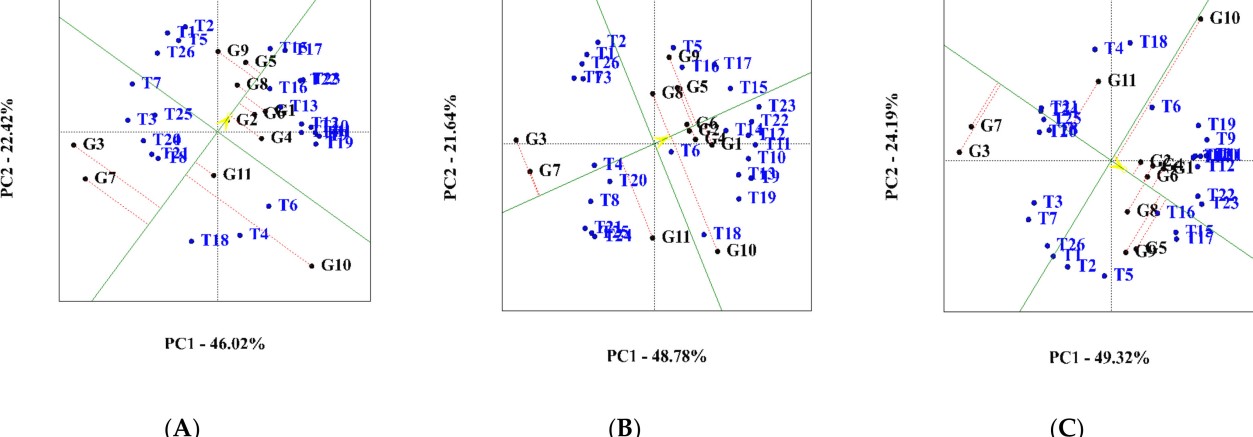

**Figure 4.** Ranking of genotypes in terms of stability in yielding. (**A**): First cropping year, (**B**): second cropping year, (**C**): average of two cropping years. G1: Argos, G2: Baron F1, G3: BATEM Tatlı, G4: Challenger, G5: Febris, G6: Khan F1, G7: Kompozit Şeker, G8: Overland, G9: SHY1036, G10: Signet, and G11: Tanem F1. T1: DS, T2: DM, T3: PH, T4: NTP, T5: NLP, T6: NEP, T7: FEH, T8: NPH, T9: EY, T10: NME, T11: YME, T12: EL, T13: ED, T14: EW, T15: NKRE, T16: NKR, T17: NKE, T18: TKW, T19: FKY, T20: GCP, T21: SSCH, T22: GMCF, T23: GMCR, T24: SSCF, T25: SSCR, and T26: GMY.

*3.4. Ranking of Genotypes Based on Ideal Genotype*

Figure 5 shows genotypes ranking according to superlative genotype which is why linear coordinates from the origin are linked to the point of the averages, continuing to the sides. The superlative genotype is the genotype inclined to the positive (+ve) end of the axis, and its vertical distance from this line was the most insignificant value. In Figure 5, the superlative point of the center of the circle was the center, which is determined by an arrow. Other genotypes are ranked based on this point. As long as the genotype arrow is shorter, the genotype is superior. Based on the obtained results of the superlative genotype diagram, Febris, Overland, and SHY1036 were the nearest genotypes to the positive (+ve) end in the year one cropping season, and Signet, BATEM Tatlı, and Kompozit Seker were the farthest genotypes from the positive (+ve) end. Therefore, they are identified as unfavorable genotypes (Figure 5A). The genotypes are ranked as follows: Febris > Overland > SHY1036 > Khan F1 > Argos > Baron F1 > Challenger > Tanem F1 > Signet > BATEM Tatlı > Kompozit Seker. Based on the obtained data results of the biplot of the year two cropping season, Khan F1 was identified as the superior genotype, and BATEM Tatlı was the non-favorable genotype. In year two cropping season genotypes were ranked as follows: Khan F1 > Baron F1 > Challenger > Argos > Febris > SHY1036 > Overland > Signet > Tanem F1 > Kompozit Seker > BATEM Tatlı (Figure 5B). The biplot data results, in the average of year one and year two cultivation seasons, show that Khan F1 is the superior genotype and BATEM Tatlı is the non-favorable genotype because the desire for sweet corn genotypes based on it is the distance from the assumed ideal genotypes. The superior genotypes are ranked as follows: Khan F1 > Argos> Overland > Challenger > Baron F1 > Febris > SHY1036 > Tanem F1 > Signet > Kompozit Seker > BATEM Tatlı (Figure 5C). The circle indicated by the arrow represents the mean of different traits. Small angles between the environment and the line that passes through the mean of the environments indicate that the environment is representative, and the higher the discrimination ability, the larger the vector of each environment [35]. The concentric circle's yield is highest for genotypes close to its center and lowest for genotypes farther from it. According to two cropping seasons results, Khan F1 was identified as the superlative genotype, and Kompozit Seker was identified as a non-desirable genotype. In all test environments, the "ideal" genotype would perform best and exhibit high stability [13]. The longest vector in the principal components 1 and 2 without projections, symbolized graphically by the arrow in the center of the concentric circles, defines the "ideal" genotype [17]. The GGE biplot analysis is effective because it enables the prediction of the genotype's mean yield in a particular environment and contributes to the recommendation of genotypes that are more stable and adapted for the region of interest [37]. Sharma et al. [38] used the GGE biplot methodology to identify winter wheat genotypes with high yield and stability; they found that within the 25 genotypes provided by the international program of Simit in Central Asia and West Asia (IWWIP) for the winter breeding program, five of them were identified as high-yield with sustainable stability. The GGE biplot analysis aids in understanding how much the first two PCs contribute to overall variation. In our study, the two-year mean score revealed that PC1 had a higher score (49.32%) than PC2, which had a lower score (24.19%). Yan and Rajcan [17] and Crevelari et al. [39] support these findings. To express a higher ability to represent all other environments, the higher PC1 score must also have a null PC2 score and a higher genotype discrimination capacity. The GGE biplot analysis aids in understanding how much the first two PCs contribute to overall variation. If the first two PCs can explain more than 60% and 10% of the total variation in the GE data, respectively, then the biplot can be said to effectively explain that variation [40]. Our results are supported by Choudhary et al. [41] and Akinyosoye [6].

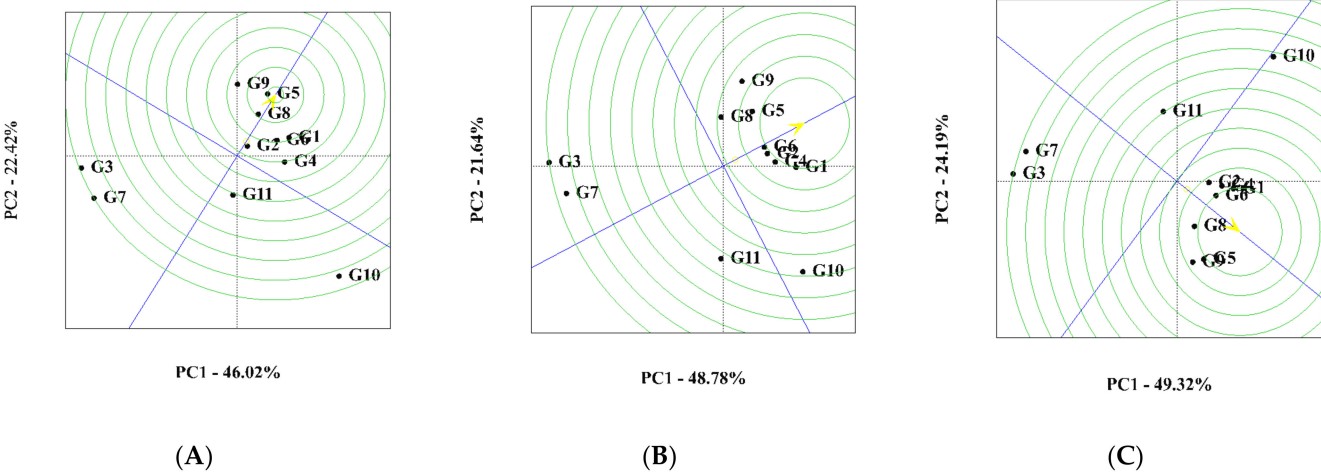

**Figure 5.** Ranking of genotypes based on ideal genotype. (**A**): First cropping year, (**B**): second cropping year, (**C**): average of two cropping years. G1: Argos, G2: Baron F1, G3: BATEM Tatlı, G4: Challenger, G5: Febris, G6: Khan F1, G7: Kompozit Şeker, G8: Overland, G9: SHY1036, G10: Signet, and G11: Tanem F1. T1: DS, T2: DM, T3: PH, T4: NTP, T5: NLP, T6: NEP, T7: FEH, T8: NPH, T9: EY, T10: NME, T11: YME, T12: EL, T13: ED, T14: EW, T15: NKRE, T16: NKR, T17: NKE, T18: TKW, T19: FKY, T20: GCP, T21: SSCH, T22: GMCF, T23: GMCR, T24: SSCF, T25: SSCR, and T26: GMY.

### 3.5. Grouping of Genotypes Diagram

The genotype ranking diagram estimates the genotypes according to the sustainable stability and yield performance in different characteristics and ranks the genotypes according to those characteristics (Figure 6). According to the grouping diagram, for year one cropping conditions, eight groups were formed concerning yield and desirability in all characteristics. The first and second groups include BATEM Tatlı and Kompozit Seker. The third group includes the Tanem F1 genotype, and the fourth group includes the Signet genotype. The fifth group includes the SHY1036, Febris, and Overland genotypes, and the sixth group includes the SHY1036, Febris, Overland, Baron F1, Khan F1, Argos, and Challenger genotypes (Figure 6A). The seventh group includes the Febris, Overland, Baron F1, Khan F1, Argos, and Challenger genotypes and the eighth group includes Baron F1, Khan F1, Argos, and Challenger genotypes (Figure 6A). according to the ranking diagram in the year two cropping conditions, eight groups were formed concerning high yield and desirability in all characteristics. The first and second groups include BATEM Tatlı and Kompozit Seker, the third group includes the Tanem F1 genotype, and the fourth group includes the Signet genotype. The fifth group includes the SHY1036, Febris, and Overland genotypes, and the sixth group includes the SHY1036, Febris, Overland, and Khan F1 genotypes (Figure 6B). The seventh group includes SHY1036 and Overland genotypes, and the eighth group includes the Overland, Febris, Khan F1, Baron F1, Challenger, and Argos genotypes (Figure 6B). According to the grouping diagram in combined cropping year conditions, seven groups were formed concerning high yield and desirability in all characteristics (Figure 6B). The first and second groups include BATEM Tatlı and Kompozit Seker, the third group includes the Tanem F1 genotype, and the fourth group includes the Signet genotype (Figure 6B). The fifth group includes the SHY1036, Febris, Overland, Khan F1, Challenger, and Argos genotypes, and the sixth group includes the Overland, Khan F1, Challenger, and Argos genotypes (Figure 6C). The seventh group includes the Overland, Khan F1, Challenger, and Argos genotypes. (Figure 6C). Examining the diagrams of first, second, and combined cropping years conditions, except the Signet and Tanem F1 genotypes, all genotypes were in the same group in both cropping conditions, identifying the sustainable stability of these genotypes regarding the studied characteristics in both cropping years and combined. Finally, in general, it can be stated that the interaction of the groups shows the similarity of the genotypes of the group in response to the traits evaluated in the experiment. Shujaei et al. [28] used genotype grouping in the study of the

interaction of genotype and traits (GT) as well as the genotype × trait × yield (GYD) and concluded that the groups that interact with members of other groups have similarities in term of the reaction to traits.

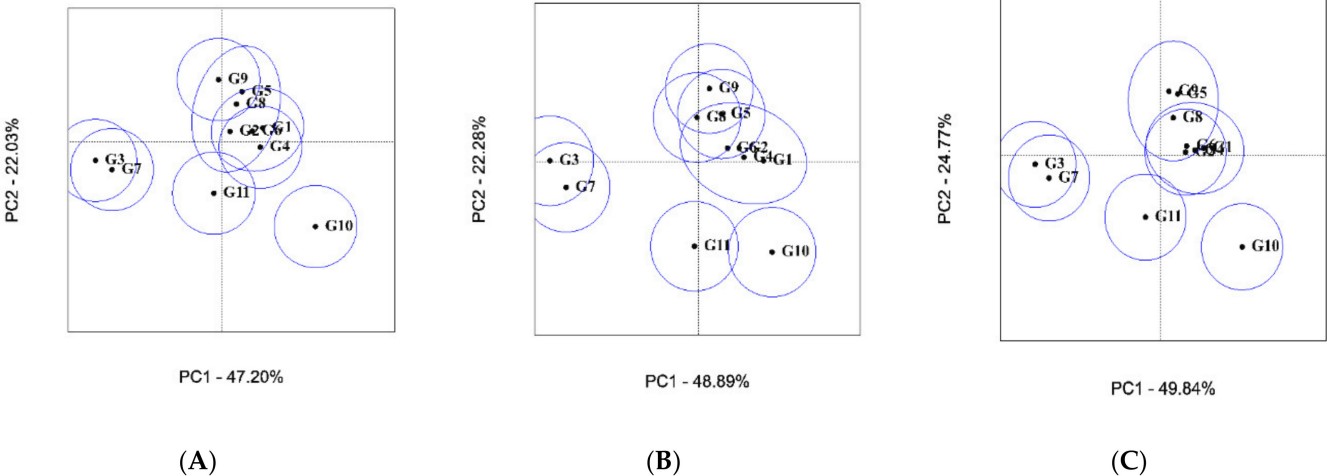

**Figure 6.** Grouping of genotypes based on traits. (**A**): First year, (**B**): second year, (**C**): average of two years. G1: Argos, G2: Baron F1, G3: BATEM Tatlı, G4: Challenger, G5: Febris, G6: Khan F1, G7: Kompozit Şeker, G8: Overland, G9: SHY1036, G10: Signet, and G11: Tanem F1. T1: DS, T2: DM, T3: PH, T4: NTP, T5: NLP, T6: NEP, T7: FEH, T8: NPH, T9: EY, T10: NME, T11: YME, T12: EL, T13: ED, T14: EW, T15: NKRE, T16: NKR, T17: NKE, T18: TKW, T19: FKY, T20: GCP, T21: SSCH, T22: GMCF, T23: GMCR, T24: SSCF, T25: SSCR, and T26: GMY.

*3.6. Analysis of Correlations between Traits*

To examine the correlation of the traits of this study, a diagram analysis of the correlation among the characteristics was applied (Figure 7). In this cosine biplot diagram, the angle between the characteristic vectors showed the intensity of the correlation among the characteristics. Assume that the angle between the vectors stands < 90°. In that circumstance, the correlation is equal to a positive one (+1). If the angle between the vectors of the attributes is equal to 90°, the correlation between the vectors of the attributes is zero (0). If the angle among the vectors is equal to 180°, the correlation is negative one (−1) [26]. According to the diagram acquired for the year one cropping season (Figure 7A), the yield of marketable ears per hectare of YME, EL, NME, FKY, EW, ED, NKR, NEP, NME, and (NKRE) together; the number of grains per ear of NKE and EL together; the PH characteristic of kernel length; and finally, the kernel yield characteristics, kernel length, kernel thickness, and number of tillers per plant of NTP showed a positive (+ve) and significant correlation. Based on the graph acquired under the second cropping season, YME, NME, ED, EY, NEP, FKY, TKW, ea EL, EW, NKRE, NKE, NKR, and (NLP) displayed a positive (+ve) and significant correlation (Figure 7B). Additionally, according to the graph obtained from the combined cropping seasons; YME showed a positive (+ve) and significant correlation with EY, FKY, NEP, TKW, EL, NKR, NKRE and NLP (Figure 7C). The 180° angles between NEP and GMY and between FKY, PH, and ED with NPH displayed a significant negative (−ve) correlation between these two characteristics (Figure 7A). According to the diagram obtained from the year two cropping season, there were positive (+ve) correlations between TKW and GMY and between fr FKY and fi FEH (Figure 7B). When we looked at the correlation result of the combined cropping years, it showed that there is a negative (−ve) correlation between FKY and DM. A negative (−ve) correlation was found between kernel yield and kernel diameter, as shown by Farajzadeh et al. [42]. In this research study, for kernel yield and its components of the 11 sweet corn genotypes, a positive (+ve) and significant correlation was determined in the number of kernels per row, number of kernels per ear, and the ear length with kernel yield [43].The correlation coefficient was obtained based on four traits: the yield of marketable ears per hectare (YME), fresh kernel yield

per hectare (FKY), crude protein (GCP), and the total soluble solids at harvest (SSCH). According to the four characteristics, correlation results showed a positive (+ve) correlation of YME with EY, NME, EL, ED, EW, NKRE, NKR, FKY, GMCF, GMCR, and (SSCF), while it showed a negative (−ve) correlation with DS, DM, PH, FEH, NPH, GCP, SSCH, SSCR, and GMY (Table 3). In this research study of kernel yield and its components of the 11 sweet corn genotypes, a positive (+ve) and significant correlation was determined in the number of grains per row, the number of grains per ear, and ear length with kernel yield [24]. In terms of the FKY, the correlation result demonstrated that there is a positive relationship between FKY and EY, NME, YME, EL, ED, EW, NKRE, NKE, SSCF, GMCF, GMCR, and SSCR, whereas it was negatively (−ve) correlated with DS, DM, PH, FEH, GCP, SSCH and GMY (Table 3). Moreover, there is a positive (+ve) correlation of GCP with DS, PH, FEH, SSCH, and (SSCR) and a negative (−ve) correlation with NLP, NEP, EY, NME, YME, EL ED, NKR, NKE, FKY, GMCF, GMCR, and SSCF (Table 3). Additionally; Mousavi et al. [43] reported a positive and significant correlation between kernel yield traits and the number of grains per row. Moreover, SSCH was positively (+ve) correlated with NTP, FEH, GCP, SSCR, and a negative (−ve) correlation was found between SSCH and NLP, NEP, EY, NME, YME, EL, ED, EW, NKRE, NKR, NKE, FKY and SSCF (Table 3 and Figure 8). In their report, Saleh et al. [44] confirmed a significant negative (−ve) correlation between total soluble solids at harvest and ear yield, ear length, ear diameter, and the number of grains per row and kernel yield in the plot. In another study conducted by Kashiani and Saleh [45], they found a significant negative (−ve) correlation of total soluble solids (TSS) with ear diameter, number of grains per ear, and the number of kernel rows per ear.

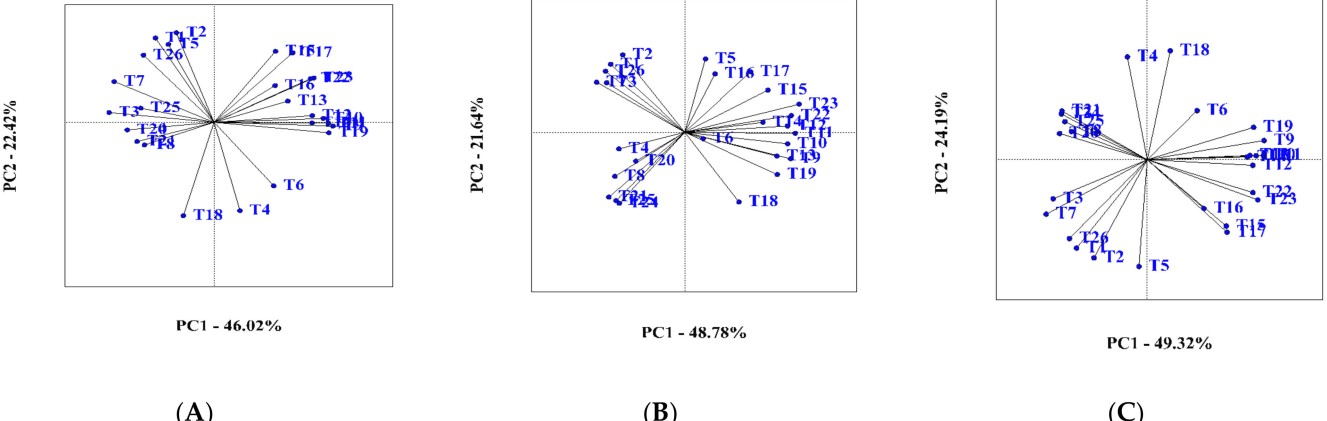

**Figure 7.** Analysis of correlations between traits. (**A**): First cropping year, (**B**): second cropping year, (**C**): average of two cropping years. G1: Argos, G2: Baron F1, G3: BATEM Tatlı, G4: Challenger, G5: Febris, G6: Khan F1, G7: Kompozit Şeker, G8: Overland, G9: SHY1036, G10: Signet, and G11: Tanem F1. T1: DS, T2: DM, T3: PH, T4: NTP, T5: NLP, T6: NEP, T7: FEH, T8: NPH, T9: EY, T10: NME, T11: YME, T12: EL, T13: ED, T14: EW, T15: NKRE, T16: NKR, T17: NKE, T18: TKW, T19: FKY, T20: GCP, T21: SSCH, T22: GMCF, T23: GMCR, T24: SSCF, T25: SSCR, and T26: GMY.

**Table 3.** Correlation coefficients among different growth, yield, and some quality traits of sweet corn genotypes.

| Traits | DS | DM | PH | NTP | NLP | NEP | FEH | NPH | EY | NME | YME | EL | ED | EW | NKRE | NKR | NKE | TKW | FKY | GCP | SSCH | GMCF | GMCR | SSCF | SSCR |
|---|---|---|---|---|---|---|---|---|---|---|---|---|---|---|---|---|---|---|---|---|---|---|---|---|---|
| DM [2] | 0.887 *** [1] | | | | | | | | | | | | | | | | | | | | | | | | |
| PH | 0.526 ** | 0.415 ** | | | | | | | | | | | | | | | | | | | | | | | |
| NTP | −0.233 | −0.326 ** | 0.046 | | | | | | | | | | | | | | | | | | | | | | |
| NLP | 0.299 * | 0.382 ** | 0.038 | −0.596 ** | | | | | | | | | | | | | | | | | | | | | |
| NEP | −0.420 ** | −0.458 ** | −0.190 | 0.298 * | −0.260 * | | | | | | | | | | | | | | | | | | | | |
| FEH | 0.643 ** | 0.505 ** | 0.750 ** | −0.100 | −0.013 | −0.355 ** | | | | | | | | | | | | | | | | | | | |
| NPH | 0.116 | 0.132 | 0.083 | −0.062 | 0.088 | −0.089 | 0.065 | | | | | | | | | | | | | | | | | | |
| EY | −0.618 ** | −0.475 ** | −0.509 ** | −0.087 | −0.158 | 0.246 * | −0.584 ** | −0.301 * | | | | | | | | | | | | | | | | | |
| NME | −0.441 ** | −0.308 * | −0.638 ** | −0.098 | −0.009 | 0.111 | −0.605 ** | −0.244 * | 0.809 ** | | | | | | | | | | | | | | | | |
| YME | −0.472 ** | −0.318 * | −0.659 ** | −0.055 | −0.066 | 0.204 | −0.653 ** | −0.323 ** | 0.840 ** | 0.931 ** | | | | | | | | | | | | | | | |
| EL | −0.359 ** | −0.197 | −0.457 ** | −0.128 | 0.056 | 0.086 | −0.524 ** | −0.381 ** | 0.759 ** | 0.781 ** | 0.833 ** | | | | | | | | | | | | | | |
| ED | −0.399 ** | −0.279 * | −0.411 ** | −0.241 | 0.049 | 0.309 * | −0.426 ** | −0.334 ** | 0.586 ** | 0.447 ** | 0.507 ** | 0.465 ** | | | | | | | | | | | | | |
| EW | −0.361 ** | −0.233 | −0.416 ** | −0.050 | −0.108 | 0.282 * | −0.402 ** | −0.300 * | 0.502 ** | 0.403 ** | 0.317 * | 0.583 ** | 0.498 ** | | | | | | | | | | | | |
| NKRE | −0.065 | 0.067 | −0.220 | −0.466 ** | 0.278 * | 0.074 | −0.101 | −0.334 ** | 0.477 ** | 0.442 ** | 0.325 ** | 0.588 ** | 0.526 ** | 0.465 ** | | | | | | | | | | | |
| NKR | 0.115 | 0.239 | −0.291 * | −0.070 | 0.263 * | 0.052 | −0.322 ** | −0.199 | 0.228 | 0.341 ** | 0.396 ** | 0.513 ** | 0.203 | 0.225 | 0.144 | | | | | | | | | | |
| NKE | 0.078 | 0.214 | −0.295 * | −0.347 ** | 0.339 ** | 0.123 | −0.243 * | −0.308 * | 0.455 ** | 0.488 ** | 0.536 ** | 0.515 ** | 0.455 ** | 0.396 ** | 0.744 ** | 0.738 ** | | | | | | | | | |
| TKW | −0.562 ** | −0.516 ** | −0.199 | 0.100 | −0.281 * | 0.158 | −0.238 | −0.059 | 0.218 | 0.140 | 0.152 | 0.192 | 0.318 ** | 0.135 | −0.253 * | −0.192 | −0.355 ** | | | | | | | | |
| FKY | −0.600 ** | −0.462 ** | −0.655 ** | −0.020 | −0.170 | 0.233 | −0.619 ** | −0.229 | 0.834 ** | 0.851 ** | 0.864 ** | 0.694 ** | 0.421 ** | 0.436 ** | 0.357 ** | 0.231 | 0.386 ** | 0.248 * | | | | | | | |
| GCP | 0.261 * | 0.050 | 0.336 ** | 0.205 | −0.320 ** | −0.322 ** | 0.463 ** | −0.307 * | −0.249 * | −0.306 * | −0.255 * | −0.406 ** | −0.335 ** | −0.422 ** | −0.248 * | −0.393 ** | 0.035 | −0.246 * | | | | | | | |
| SSCH | 0.130 | −0.038 | 0.235 | 0.272 * | −0.389 ** | 0.207 | −0.025 | −0.388 ** | 0.207 | −0.354 ** | −0.432 ** | −0.452 ** | −0.433 ** | −0.402 ** | −0.423 ** | −0.351 ** | −0.491 ** | 0.112 | −0.253 * | 0.770 ** | | | | | |
| GMCF | −0.206 | −0.028 | −0.524 ** | −0.306 * | 0.207 | −0.025 | −0.591 ** | −0.311 * | 0.721 ** | 0.665 ** | 0.739 ** | 0.763 ** | 0.654 ** | 0.444 ** | 0.533 ** | 0.420 ** | 0.600 ** | −0.103 | 0.639 ** | −0.314 * | −0.420 ** | | | | |
| GMCR | −0.203 | −0.026 | −0.449 ** | −0.285 * | 0.223 | −0.025 | −0.523 ** | −0.352 ** | 0.665 ** | 0.739 ** | 0.763 ** | 0.718 ** | 0.654 ** | 0.497 ** | 0.553 ** | 0.420 ** | 0.600 ** | −0.158 | 0.714 ** | −0.342 ** | −0.465 ** | 0.909 ** | | | |
| SSCF | −0.206 | −0.028 | −0.524 ** | −0.306 * | 0.207 | −0.025 | −0.591 ** | −0.311 * | 0.721 ** | 0.771 ** | 0.771 ** | 0.804 ** | 0.718 ** | 0.529 ** | 0.477 ** | 0.533 ** | 0.520 ** | −0.103 | 0.714 ** | −0.211 | 0.571 ** | 0.733 ** | −0.401 ** | | |
| SSCR | 0.215 | 0.087 | 0.134 | 0.164 | −0.226 | 0.207 | −0.352 ** | 0.337 ** | 0.243 * | −0.334 ** | −0.295 * | −0.391 ** | −0.379 ** | −0.351 ** | −0.330 ** | −0.361 ** | −0.294 * | −0.409 ** | −0.045 | −0.211 | −0.326 ** | −0.293 * | | | |
| GMY | 0.624 ** | 0.088 | 0.601 ** | 0.606 ** | −0.320 ** | 0.336 ** | −0.205 | 0.476 ** | 0.349 ** | −0.422 ** | −0.480 ** | −0.483 ** | −0.405 ** | −0.229 | −0.248 * | 0.057 | −0.093 | 0.023 | −0.468 ** | −0.605 ** | 0.104 | 0.008 | −0.241 | −0.185 | 0.088 |

[1] The explanation of *, **, *** are significant at 0.05, 0.01, and 0.001 respectively. [2] DS: days to silking, DM: days to maturity, PH: plant height, NTP: number of tillers per plant, NLP: number of leaves per plant, NEP: number of ears per plant, FEH: first ear height, NPH: number of plants per hectare, EY: ear yield, NME: number of marketable ears per hectare, YME: the yield of the marketable ear, EL: ear length, ED: ear diameter, EW: ear weight, NKRE: number of kernel rows per ear, NKR: number of kernels per row, NKE: number of kernels per ear, T18: TKW:1000-kernel weight, FKY: fresh kernel yield, GCP: grain protein content, SSCH: the total soluble solid content was measured by using a digital hand refractometer at harvest, GMCF: grain moisture content was determined as a percentage at seven days post-harvest in the field, GMCR: grain moisture content was determined as a percentage at seven days post-harvest in the refrigerator, SSCF: ears with harvest date delayed one week at the field, SSCR: seven days post-harvest for ears with husk which were stored in the refrigerator, and GMY: green mass yield.

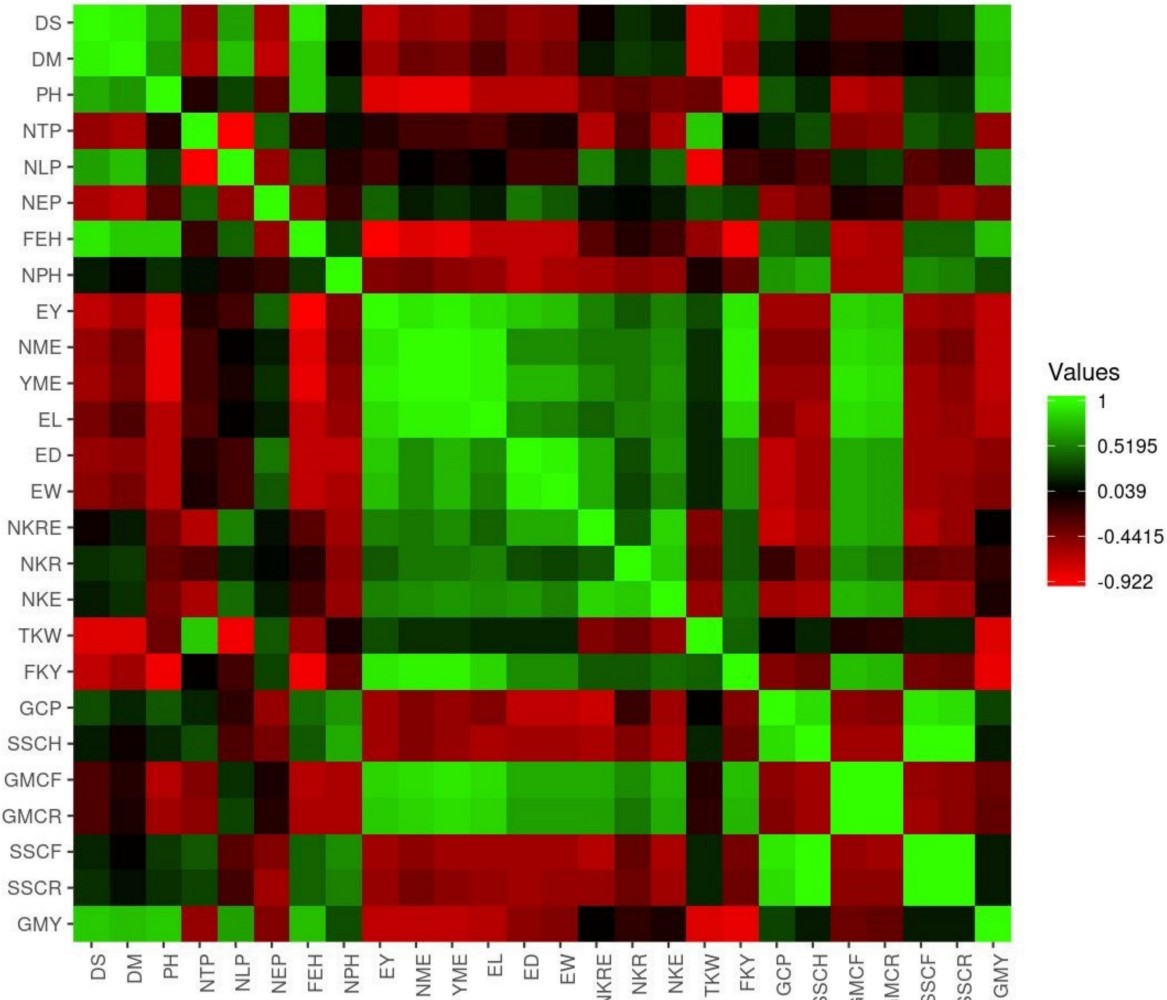

**Figure 8.** Pairwise among different growth, yield, and some quality traits of sweet corn genotypes. DS: days to silking, DM: days to maturity, PH: plant height, NTP: number of tillers per plant, NLP: number of leaves per plant, NEP: number of ears per plant, FEH: first ear height, NPH: number of plants per hectare, EY: ear yield, NME: number of marketable ears per hectare, YME: the yield of the marketable ear, EL: ear length, ED: ear diameter, EW: ear weight, NKRE: number of kernel rows per ear, NKR: number of kernels per row, NKE: number of kernels per ear, T18: TKW:1000-kernel weight, FKY: fresh kernel yield, GCP: grain protein content, SSCH: the total soluble solid content was measured by using a digital hand refractometer at harvest, GMCF: grain moisture content was determined as a percentage at seven days post-harvest in the field, GMCR: grain moisture content was determined as a percentage at seven days post-harvest in the refrigerator, SSCF: ears with harvest date delayed one week at the field, SSCR: seven days post-harvest for ears with husk which were stored in the refrigerator, and GMY: green mass yield.

*3.7. Different Traits Profiles of Sweet Corn Genotypes*

The results of hierarchical cluster analysis (Figure 9) for the 11 sweet corn genotypes and the heat map were developed based on different growth, yield, and some quality traits. Heat mapper is simple in use and allows dynamic and flexible display of a correlation plot in combination with sample characteristics. In addition to the overall reaction, patterns for different sweet corn genotypes and traits were noted, and the responses of the various clusters could be and visualized. In this way, it became evident that differences existed in the individual clusters with regards to the different traits. Based on the heatmap dendrogram showing the context of the differential extent of trait association, all genotypes were indicated into two clusters. These clusters were also clearly separated in most cases. Of the two

clusters, Cluster 1 included the Febris and SHY1036 genotypes. Cluster 2 was the greatest and was more pronounced. This cluster was comprised of two subclusters. Subcluster, I contained Signet, Challenger, and Tanem F1, whereas subcluster II included Kompozit Seker, Argos, Overland, Khan F1, Baron F1, and BATEM Tatlı (Figure 8). Heatmap analysis from various traits in sweet corn genotypes showed variability was found in the entire gene pool for all traits studied. The highest genetic distance was shown between BATEM Tatlı and Febris genotypes. Differential responses of sweet corn genotypes in terms of quality and agronomic traits confirmed variations in their gene architectures. In the study conducted by Shujaei et al. [28], the genotypes were grouped by traits using a heatmap, and the results of this study were consistent with the results of their study.

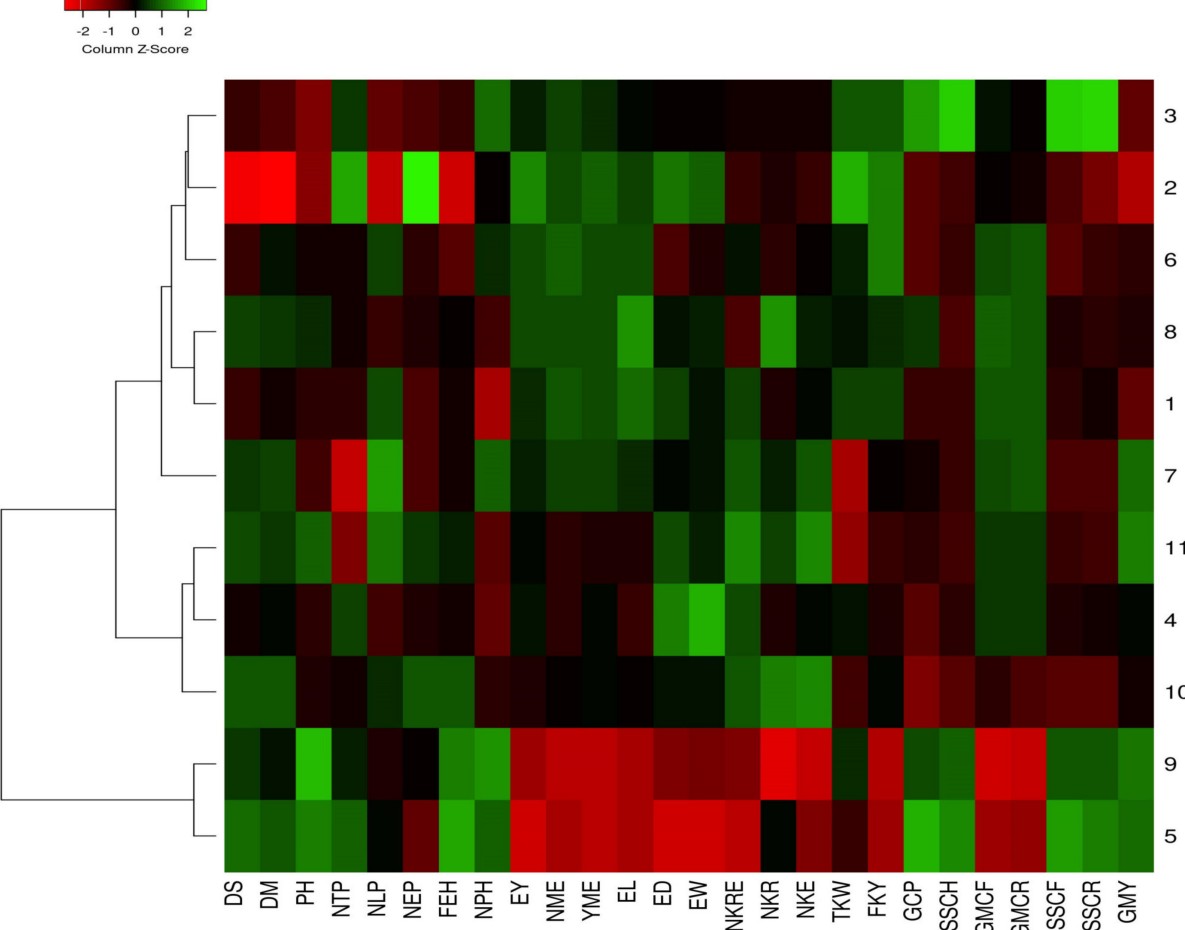

**Figure 9.** Heatmap dendrogram dividing sweet corn genotypes into different clusters based on different growth, yield, and some quality traits. DS: the days to silking, DM: days to maturity, PH: plant height, NTP: number of tillers per plant, NLP: number of leaves per plant, NEP: number of ears per plant, FEH: first ear height, NPH: number of plants per hectare, EY: ear yield, NME: number of marketable ears per hectare, YME: the yield of the marketable ear, EL: ear length, ED: ear diameter, EW: ear weight, NKRE: number of kernel rows per ear, NKR: number of kernels per row, NKE: number of kernels per ear, T18: TKW:1000-kernel weight, FKY: fresh kernel yield, GCP: grain protein content, SSCH: the total soluble solid content was measured by using a digital hand refractometer at harvest, GMCF: grain moisture content was determined as a percentage at seven days post-harvest in the field, GMCR: grain moisture content was determined as a percentage at seven days post-harvest in the refrigerator, SSCF: ears with harvest date delayed one week at the field, SSCR: seven days post-harvest for ears with husk which were stored in the refrigerator, and GMY: green mass yield. 1: Argos, 2: Baron F1, 3: BATEM Tatlı, 4: Challenger, 5: Febris, 6: Khan F1, 7: Kompozit Şeker, 8: Overland, 9: SHY1036, 10: Signet, and 11: Tanem F1.

## 4. Conclusions

In this research, the GT biplot method was used, which is derived from the GGE biplot method for selecting suitable genotypes of sweet corn. Analyses carried out showed that Khan F1 was named the stable genotype. Additionally, based on year one cropping, year two cropping, and the average of the two cropping seasons, it can be stated that Khan F1 is the highest-yielding genotype. Our results showed that the summation of the first two and second main components was accountable for 73.51% of combined cropping years of the sweet corn performance difference, indicating the high relative validity of the biplot diagram obtained from this research study. Based on the heatmap dendrogram the context of the differential extent of trait association, all genotypes are indicated into two clusters. The highest genetic distance was shown between BATEM Tatlı and Febris genotypes. Heatmap analysis from various traits in sweet corn genotypes showed variability was found in the entire gene pool for all traits studied. Our results provide helpful information about the sweet corn genotypes and environments for future breeding programs.

**Author Contributions:** Conceptualization, A.A.L.S. and A.Ö.; methodology, A.A.L.S., A.Ö. and G.N.; software, K.H. and A.T.; validation, A.T., G.N., P.S., T.W. and M.P.; formal analysis, K.H., A.T., A.O., G.N., P.S., T.W. and M.P.; investigation, K.H. and A.T.; resources, K.H., A.T. and A.Ö.; data curation, A.Ö., K.H. and A.T.; writing—original draft preparation, A.A.L.S., K.H., A.T., G.N., P.S., T.W. and M.P.; writing—review and editing, K.H., A.T., G.N., P.S., T.W. and M.P.; visualization, A.Ö., K.H. and A.T.; supervision, A.Ö., K.H. and G.N.; project administration, A.Ö.; funding acquisition, A.Ö.; All authors have read and agreed to the published version of the manuscript.

**Funding:** This research received no external funding.

**Data Availability Statement:** All data supporting the conclusions of this article are included in this article.

**Acknowledgments:** We thank the anonymous reviewers and editors for their constructive comments on this manuscript. Additionally, we acknowledged the support of the "The Atatürk University, Scientific Research Projects Commission" (Project number: FYL-2018-6631).

**Conflicts of Interest:** The authors declare no conflict of interest.

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
