# Peer review of "Genotype–Trait (GT) Biplot Analysis for Yield and Quality Stability in Some Sweet Corn (Zea mays L. saccharata Sturt.) Genotypes"

_agronomy, doi:10.3390/agronomy13061538_

Round 1

Reviewer 1 Report

Line51-52: add references to the data

Lines53-54: rephase

Line 59: ; ->.

Line61-62: rephase

Line 75: exit -> exits

Line 105: , ->.

Line 116: this study's research -> this study

Line 438: a -> the

The English writing needs to be improved throughout the whole paper. Please look for help from the writing center for suggestions.

Author Response

Responses to Comments of Reviewer 1

General Response:

Dear reviewer; According to the valuable suggestions of the commentators, the comments of Reviewer 1 on the manuscript are highlighted in red. Thank you for giving us the chance to review our manuscript. In the table below, we tried to respond to the suggestions and comments of all the referees in the best way possible.

Comments Reviewer 1

The present study is very impressive. The authors took great efforts in understanding and establishing the sweet corn genotypes yield and stability which would be helpful for future breeding research. I have only few minor points that needs to be addressed.

Comment

Response

1.     Line51-52: add references to the data

The requested edit was done.

2.     Lines53-54: rephase

The requested edit was done.

3.     Line 59: ; ->.

The requested edit was done.

4.     Line61-62: rephase

The requested edit was done.

5.     Line 75: exit -> exits

The requested edit was done.

6.     Line 105: , ->.

The requested edit was done.

7.     Line 116: this study's research -> this study

The requested edit was done.

8.     Line 438: a -> the

The requested edit was done.

9.     The English writing needs to be improved throughout the whole paper. Please look for help from the writing center for suggestions.

To polish the language and elimite errors, we have revised the text by a native person. All changes are shown by Track-changes.

Reviewer 2 Report

In this manuscript, the authors compare the 11 sweet corn genotypes for two crop years in Erzurum Turkey using RCBD experimental design  and then find out the highest yielding and the most suitable sweet corn genotype for Erzurum ecological conditions.
In some places, some changes are needed that I mentioned below.

1-The manuscript should be revised carefully for English writing.

2-Please include some available literature in the introduction about the utilization of GGE Biplot analysis in sweet corn and other crops.

3-My personal suggestion is to conduct multiple experiments in multiple places in Erzurum to increase the reliability of the results.

4-In the Materials and methods section, 11 sweet corn information and planting location information should be shown in table form.

5-Personally, Perhaps the graphics (Figure 1,2,3,4,5) could be more aesthetically pleasing, such as in font selection, color depth, and sharpness, because the image is a little distorted.

6-Although the abbreviations in the heatmap are explained in the Materials and Methods, Perhaps it is more appropriate to add them again in the notes (Figure 6,7). Whta's more,  It is recommended that the horizontal and vertical fonts in the graphic should be consistent with the original text.

7-In the results and discussion section, the contents discussed are relatively small, so it is necessary to strengthen the corresponding literature reading, increase the comparison and discussion between previous studies and this paper, and supplement and enrich the contents of this part.

The manuscript should be revised carefully for English writing.

Author Response

Responses to Comments of Reviewer 2

General Response:

Dear editor; According to the valuable suggestions of the commentators, the comments of Reviewer 2 on the manuscript are highlighted in green. Thank you for giving us the chance to review our manuscript. In the table below, we tried to respond to the suggestions and comments of all the referees in the best way possible.

Comments Reviewer 2

GENERAL COMMENT

Response

1-The manuscript should be revised carefully for English writing.

It was reviewed again.

2-Please include some available literature in the introduction about the utilization of GGE Biplot analysis in sweet corn and other crop

Added requested items to the introduction.

3-My personal suggestion is to conduct multiple experiments in multiple places in Erzurum to increase the reliability of the results.

It will definitely be considered in other experiments.

4-In the Materials and methods section, 11 sweet corn information and planting location information should be shown in table form.

The characteristics of the soil of the region as well as the climate and rainfall of the region were added.

5-Personally, Perhaps the graphics (Figure 1,2,3,4,5) could be more aesthetically pleasing, such as in font selection, color depth, and sharpness, because the image is a little distorted.

The changes have been applied.

6-Although the abbreviations in the heatmap are explained in the Materials and Methods, Perhaps it is more appropriate to add them again in the notes (Figure 6,7). Whta's more,  It is recommended that the horizontal and vertical fonts in the graphic should be consistent with the original text.

The changes have been applied.

7-In the results and discussion section, the contents discussed are relatively small, so it is necessary to strengthen the corresponding literature reading, increase the comparison and discussion between previous studies and this paper, and supplement and enrich the contents of this part.

Added requested items to the results.

Reviewer 3 Report

The manuscript entitled “Assessment of Sweet Corn (Zea mays L. saccharata Sturt.) Genotypes for Yield and Quality Stability Using GGE Biplot Methods”, concerns a field experiment aiming at evaluating the genotype x environment interaction of sweet corn in Turkey. The subject of the manuscript falls with the general scope of the Journal. However, with regret it shows severe and important gaps and weaknesses.

GENERAL COMMENTS

·       The level of English language, grammar and style is insufficient. It needs to be enhanced with the help of a mother-tongue proof-reader;

·       Size and dimensions of figures 1-5 do not fit properly with the manuscript. Please re-dimension them

·       Materials and methods are almost absent. Please divide them into sub-paragraphs and provide complete information

·       The novelty level seems to be low since any difference is provided with references 17 and 18. Moreover, almost any information about M&Ms is provided here, remanding to those references

·       In such kind of studies, the environment is generally the combination of location x year. An initial ANOVA table is of key importance to dissect the effect of genotype, environment and their interaction. It is also important dissecting in supplementary materials the specific role of location and year to understand which of them contribute more to variance for the environmental effect

·       Overall, statistical analysis was therefore carried out not properly, thus making the results not adequate. Please see the specific comments

SPECIFIC COMMENTS:

·       Title: I suggest changing deleting the brackets to reduce the total length

ABSTRACT:

·       Overall, it is too long. Please summarise it to max 200 words

·       the aims are not well explained and they are confused with M&Ms. Please rephrase the aims

·       results are too long and they not highlight appropriately the main findings

·       L41-42: there are grammar mistakes

·       the conclusive remarks are redundant with the same concept repeated several times

INTRODUCTION

·       L49: please provide a reference

·       L50: please provide a reference

·       L52: please provide a reference

·       L59: please change with a full stop

·       L60: this reference is not appropriate, since it is about chickpea. Please change the reference

·       L80-87: reference too long and with grammar mistakes. Please check and rephrase it

·       L87-103: the information of this section is very redundant. Please summarises it significantly

·       L104-115: it is the same as the previous comment. I suggest changing with other information. For instance, that GGE biplot analysis is generally carried out for the evaluation of productive performances, for which many examples are available in the literature. On the contrary, GGE biplot analysis on qualitative characteristics are not common as yield. You should provide some examples for similar articles about both yield and quality characteristics.

·       For GGE biplot analysis on yield parameters many references are available. About quality characteristics, which are nor reported in this study, I suggest this recent paper: “Genotype × environment interactions of potato tuber quality characteristics by AMMI and GGE biplot analysis” (https://doi.org/10.1016/j.scienta.2022.111750)

·       L118-122: sentence not clear, please rephrase it

·       Please explain better the novelty level and the goals of this study. Specifically, what are the differences with references 17 and 18?

MATERIALS AND METHODS:

·       L128: which papers? Please cite them. And what are the differences with them?

·       The first sentence is more appropriate for the end of the introduction, in order to specify the differences with this study and highlight the novelty level (if any)

·       It is not clear which genotypes were evaluated in this study. Please provide a Table with complete information about them

·       How many locations were studied? And where?

·       What are the weather conditions during the two growing season? And what about soil characteristics?

·       Please add a Table with the locations under study, their geographical coordinates, soil characteristics and weather conditions during the two years

·       Any information about agronomic management and experimental design is provided

·       Any information in provided about how the showed variables have been obtained

·       Statistical analysis is not described and carried out properly (anything about ANOVA, software, etc.)

RESULTS: due to the improper statistical approach, they are not valid in the form they are showed. Consequently, conclusions are not supported.

·   The level of English language, grammar and style is insufficient. It needs to be enhanced with the help of a mother-tongue proof-reader

Author Response

Responses to Comments of Reviewer 3

General Response:

Dear editor; According to the valuable suggestions of the commentators, the comments of Reviewer 2 on the manuscript are highlighted in blue. Thank you for giving us the chance to review our manuscript. In the table below, we tried to respond to the suggestions and comments of all the referees in the best way possible.

Comments Reviewer 2

GENERAL COMMENT

Response

·       The level of English language, grammar and style is insufficient. It needs to be enhanced with the help of a mother-tongue proof-reader;

 To polish the language and elimite errors, we have revised the text by a native person. All changes are shown by Track-changes.

·       Size and dimensions of figures 1-5 do not fit properly with the manuscript. Please re-dimension them

 The size of the requested figures was arranged

·       Materials and methods are almost absent. Please divide them into sub-paragraphs and provide complete information

Sub-paragraphs were added to Materials and Methods

·       The novelty level seems to be low since any difference is provided with references 17 and 18. Moreover, almost any information about M&Ms is provided here, remanding to those references

This manuscript is a supplement to the aforementioned articles and in this research, graphic analysis and also the use of grouping in the form of a heatmap were used in order to check the traits and select the most suitable hybrids.

·       In such kind of studies, the environment is generally the combination of location x year. An initial ANOVA table is of key importance to dissect the effect of genotype, environment and their interaction. It is also important dissecting in supplementary materials the specific role of location and year to understand which of them contribute more to variance for the environmental effect

In this study there is an environment and the interaction effect of genotype and trait is investigated using the GGE biplot method, which is introduced as GT biplot. The results of analysis of variance were added to the article

SPECIFIC COMMENTS:

·       Title: I suggest changing deleting the brackets to reduce the total length

The change has been applied.

ABSTRACT:

·       Overall, it is too long. Please summaries it to max 200 words

The change has been applied.

·       The aims are not well explained and they are confused with M&Ms. Please rephrase the aims

The changes have been applied.

·       Results are too long and they not highlight appropriately the main findings

The changes have been applied.

·       L41-42: there are grammar mistakes

The grammar mistakes have been fixed.

The conclusive remarks are redundant with the same concept repeated several times

The changes have been applied.

INTRODUCTION

Reference added.

·       L49: please provide a reference

·       L50: please provide a reference

·       L52: please provide a reference

·       L59: please change with a full stop

The changes have been applied.

·       L60: this reference is not appropriate, since it is about chickpea. Please change the reference

The reference has been changed.

·       L80-87: reference too long and with grammar mistakes. Please check and rephrase it

The changes have been applied.

·       L87-103: the information of this section is very redundant. Please summarizes it significantly

Summarization was done.

·       L104-115: it is the same as the previous comment. I suggest changing with other information. For instance, that GGE biplot analysis is generally carried out for the evaluation of productive performances, for which many examples are available in the literature. On the contrary, GGE biplot analysis on qualitative characteristics are not common as yield. You should provide some examples for similar articles about both yield and quality characteristics.

The changes have been applied.

·       For GGE biplot analysis on yield parameters many references are available. About quality characteristics, which are nor reported in this study, I suggest this recent paper: “Genotype × environment interactions of potato tuber quality characteristics by AMMI and GGE biplot analysis” (https://doi.org/10.1016/j.scienta.2022.111750)

Added requested reference to article.

·       L118-122: sentence not clear, please rephrase it

The changes have been applied.

·       Please explain better the novelty level and the goals of this study. Specifically, what are the differences with references 17 and 18?

This manuscript is a supplement to the aforementioned articles and in this research, graphic analysis and also the use of grouping in the form of a heatmap were used in order to check the traits and select the most suitable hybrids.

MATERIALS AND METHODS:

·       L128: which papers? Please cite them. And what are the differences with them?

The changes have been applied.

·       It is not clear which genotypes were evaluated in this study. Please provide a Table with complete information about them

Tables 1 were added to the Materials and Methods section.

·       How many locations were studied? And where?

This study was conducted in one region and the experimental region is in Erzurum, Turkey

·       What are the weather conditions during the two-growing season? And what about soil characteristics?

Supplementary tables were added to the Materials and Methods section.

·       Please add a Table with the locations under study, their geographical coordinates, soil characteristics and weather conditions during the two years

Supplementary tables were added to the Materials and Methods section.

·       Any information about agronomic management and experimental design is provided

The changes have been applied.

·       Any information in provided about how they showed variables have been obtained

The changes have been applied.

·       Statistical analysis is not described and carried out properly (anything about ANOVA, software, etc.)

The changes have been applied.

RESULTS: due to the improper statistical approach, they are not valid in the form they are showed. Consequently, conclusions are not supported.

The changes have been applied.

 The level of English language, grammar and style is insufficient. It needs to be enhanced with the help of a mother-tongue proof-reader

To polish the language and elimite errors, we have revised the text by a native person. All changes are shown by Track-changes.

Round 2

Reviewer 3 Report

Dear authors, although some improvements, with regret I observed that several comments have been not addressed. 

-  the manuscript is not formatted according to the journal requirements

 - the level of English language is still insufficient

- L24: please add the full binomial name

- L27 is for M&Ms, not for abstract

- L45-48: sentence too long

- Differences with references 11 and 12 are still not explained. Also the author's response does not convinve me. In my opinion, this manuscript has a very low novelty level

- L90-120: this section is redundant, with the same concept repeated many times. It can be summarised

- L139. something is missining

- Only one location and two growing season is not acceptable for GGE biplot analysis. 

- As a consequence, being just one location, Table 2 can be directly incorporated into the text.

- Agronomic management needs to be explained, not reminded to previous publications

- the statistical analysis is still explained very poorly. Moreover, the ANOVA table as previously suggested is missing. 

- the resolution of Figures 7 and 8 is too low.

The level of English language is still insufficient. A mother-tongue or professional service is needed.

Author Response

Responses to Comments of Reviewer 3

General Response:

First of all, we thank the potential reviewer for her/his valuable time and also raised helpful comments and suggestions. In this step of revision, we have tried to respond to all comments and addressed all questions. We hope the revised version of manuscript gets positive feedback from you and will be acceptable for publication in the Agronomy journal. All revised parts have been highlighted in blue.

Sincerely,

Dr. Aras Turkoglu

Comments

Comment 1# the manuscript is not formatted according to the journal requirements.

Response to Comment 1# It was formatted according to the journal requirements.

Comment 2# the level of English language is still insufficient

Response to Comment 2# We thank the potential reviewer for highlighting these mistakes. In the revised version, we have improved the language by helping a native person (Dr. Wojciechowski).

Comment 3# -L24: please add the full binomial name.

Response to Comment 3# This sentences was re-written and it has been improved.

Comment 4# -L27 is for M&Ms, not for abstract.

Response to Comment 4# It has been removed.

Comment 5# -L45-48: sentence too long.

Response to Comment 5# This sentences was re-written.

Comment 6# - Differences with references 11 and 12 are still not explained. Also, the author's response does not convince me. In my opinion, this manuscript has a very low novelty level

Response to Comment 6# It was corrected; 

Comment 7# - L90-120: this section is redundant, with the same concept repeated many times. It can be summarized.

Response to Comment 7# This sentences was re-written.

Comment 8# - L139. something is missining.

Response to Comment 8# It has been improved.

Comment 9# - Only one location and two growing season is not acceptable for GGE biplot analysis. As a consequence, being just one location

Response to Comment 9# Dear reviewer, the genotypes used in this research are new and high-yielding genotypes in the Türkiye, and so far, no experiments have been conducted on the interaction between genotype and trait on these genotypes. In addition, the purpose of this study is;

1) Examining the effect of genotype × trait using the GT biplot method and to select several traits with high performance and success

2) To identify the genotypes that had the most desirable for different traits

3) To study the correlation between the studied traits and the relationships between them

4) Grouping of genotypes according to the studied traits.

Comment 10# - Table 2 can be directly incorporated into the text.

Response to Comment 10# This tale was added in the main text.

Comment 11# - the statistical analysis is still explained very poorly. Moreover, the ANOVA table as previously suggested is missing. 

Response to Comment 11# It was done once again and improved.

Comment 11# - the resolution of Figures 7 and 8 is too low.

Response to Comment 12# It was replaced with clear one.
